# Duality of Seasonal Effect and River Bend in Relation to Water Quality in the Chao Phraya River

**Guangwei Huang [1],\*, Han Xue [1], Huan Liu [1], Chaiwat Ekkawatpanit [2] and Thada Sukhapunnapha [3]**

[1] Graduate School of Global Environmental Studies, Sophia University, Tokyo 102-8554, Japan; xuehan@sophia.ac.jp (H.X.); liulittlehuan@gmail.com (H.L.)
[2] Department of Civil Engineering, King Mongkut's University of Technology Thonburi, Bangkok 10140, Thailand; chaiwat.ekk@kmutt.ac.th
[3] Royal Irrigation Department, Bangkok 10300, Thailand; thada999@yahoo.com
\* Correspondence: huanggwx@sophia.ac.jp; Tel.: +81-3-3238-4667

**Abstract:** The present study conducted a field survey of water quality along the Chao Phraya River during the past three years. The main objective was to better understand the spatial–temporal variations in water quality in relation to season and channel morphology. It assessed the water quality in terms of chemical parameters, bacteria, and phytoplankton. The results revealed a duality of seasonal effect for nutrients. The rainy season degraded the water quality by increasing the nutrient concentration in the waterway in the beginning, but cleaned it up by dilution in the end. However, this duality did not apply to *Escherichia coli* (*E. coli*), for which the highest level occurred during the second half of the rainy season and a sag curve variation pattern was displayed along the mainstream. Another duality found by this study is that there was no statistically significant difference in water quality in terms of chemical parameters between a river bend and the straight channel shortcutting the bend, but significant differences in the level of *E. coli* and the phytoplankton community structure were observed between the two. Of particular note, the present study revealed a coexistence of a saproxenous species (algae found in clean water) with a harmful species in the bend river reach.

**Keywords:** Chao Phraya River; water quality variation; *E. coli*; phytoplankton; river bend

## 1. Introduction

Although river water quality has been studied, either routinely by river management authorities to determine the compliance with ambient and discharge standards [1], or by researchers for various specific objectives [2,3], spatial–temporal variations in water quality in rivers remain an important area of water environmental studies as evidenced by various investigations and assessments in recent years. Bouza-Deaño et al. [4] employed the non-parametric Mann–Kendall Test to detect the trends of more than 30 physical–chemical and chemical variables in the Ebro River (Spain) over a period of 24 years, and revealed that the water quality variation over time was due to decreasing phosphate concentrations and elevated pH levels. Bu et al. [5] studied the spatial and temporal variations in water quality in the Jinshui River of the South Qinling Mountains, China for three years and identified untreated domestic sewage and agricultural runoff from the eroded land as the main causes of the pollution. Wang et al. [6] investigated the distribution of dissolved heavy metals in the Changjiang River for two years and identified two large zones mainly influenced by mineral erosion and anthropogenic action, respectively. Ren et al. [7] targeted the spatial and temporal variation of water quality in a highly artificialized urban river reach and the results showed that temporal variation is greater than spatial, and sewage discharge is the dominant factor of seasonal distribution. A study on temporal variations

in water quality in the river Rwizi within the Mbarara municipality, Uganda concluded that water pollution resulted primarily from domestic waste water, agricultural runoff, and industrial effluents [8]. Another study on the spatial and temporal variations of water quality along the Bagmati River and its tributaries in the Kathmandu valley of Nepal, reported that the water qualities upstream were increasingly affected by human sewage and chemical fertilizer, while in the urban areas downstream, the river was heavily polluted with untreated municipal sewage [9]. Chang [10] examined the spatial patterns of water quality trends for 118 sites in the Han River basin of South Korea and suggested that spatial analysis of watershed data at different scales should be a vital part of identifying the fundamental spatial–temporal distribution of water quality.

A common feature of many previous studies is that they attempted to relate the spatial and temporal patterns of water quality variation with underlying causes such as anthropogenic activities, climate change, and catchment characteristics, and focused mainly on physical–chemical and chemical variables. In comparison, the microbial communities of rivers and streams have been studied relatively rarely [11]. Moreover, phytoplankton plays a central role in the functioning of freshwater ecosystems, such as nutrient cycling, and thereby can serve as bio-indicator of environmental changes, since its assemblage is always influenced by environmental factors [12,13]. Thus, efforts to combine chemical, bacterial, and phytoplankton information will certainly deepen our understanding of water quality variation and lead to better river management.

On the other hand, river channels display widely varying characteristics in both space and time, reflecting their geographical location within a particular catchment and fluvial system. Human activity affects both water quality and river channel morphology [14]. In the past, many rivers have been straightened by cutting off meanders to speed up the drainage of water and control/limit the river bed movements [15]. Channelizing was also a way to gain land for cultivation. Therefore, channel morphology and its alteration by river engineering works can be considered a deeply relevant subject when dealing with water quality. Flow dynamics and sediment transport in bends have been extensively studied [16,17] and the importance of bends to river ecology is well recognized. In recent years, re-meandering of rivers has become a common and widely applied river rehabilitation measure [18]. In addition to improving conditions for the biological quality elements, re-meandering could also help to improve habitats for birds and mammals that prey on fish and invertebrates [19]. Although river bends have been an important research target in various fields, from river engineering and geomorphology to river ecology, the effects of bends on water quality remain insufficiently investigated. Consequently, field data of water quality variations along river bends remain limited. Lee [20] studied the water qualities of three oxbow lakes, or cut-off bends, in Malaysia, and reported that all oxbow lakes exhibited wide daily variation in dissolved oxygen and pH, which is often observed in lakes where eutrophication has occurred. A study by Xiao et al. [21] focused on the correlation between river sinuosity and self-purification capacity, and the results showed that river sinuosity has different degrees of positive correlation with the growth rate of dissolved oxygen and the reduction rates of total nitrogen, ammonia nitrogen, and total phosphorus. However, this study was based on the assumption that the concentration of pollutants decays from upstream to downstream, so that the reduction rates in different river reaches can be estimated and used to quantify the self-purification capacity of meandering reaches. Therefore, these findings are useful in helping to understand the spatial variation of water quality in some specific rivers, but not applicable to rivers where there are diffuse pollutant sources along the river course. A very recent study focusing on hyporheic exchange within the intra-meander region indicated that the meander acted as a sink for organic and inorganic carbon and iron during the extended baseflow and high-water conditions, thereby affecting river water quality [22].

Considering the complex variation in water quality across time and space, an effective management of river water quality requires two key types of information: (1) spatial and temporal variations in water quality, and (2) information about the driving factors influencing the water quality.

More research focusing on spatial and temporal variations of water quality and its relationship with river channel patterns can therefore be justified.

In the present study, the Chao Phraya River was chosen due to its persistent pollution, channel patterns, the population distribution in the watershed, and the significance of the river in Thailand. The primary objectives were: (1) to characterize the overall spatial and temporal variability of water quality in the Chao Phraya River at present, with a particular focus on water quality differences between the beginning and end of the rainy season in terms of both chemical variables and *Escherichia coli* (*E. coli*), and (2) to examine the water quality variation in relation to river bend, with an additional probe into phytoplankton communities in the river course, which has not been previously reported. A secondary objective was to provide an outlook on the relationship between the population distribution and spatial variation of water quality in the watershed, by mapping the two together.

## 2. Materials and Methods

### 2.1. Study Area

The present study conducted field surveys of water quality along the Chao Phraya River from 2016 to 2018. The Chao Phraya River is the largest river in Thailand. Its basin covers an area of 159,000 km$^2$, accounting for 30% of Thailand's land surface area, hosting 40% of the country's population, and generating 66% of the Gross Domestic Product (GDP) [23]. The Chao Phraya basin is influenced by the south western monsoon and also the north eastern monsoon, and therefore it creates three seasons in the area. The rainy season is from May to October, the cool season is from November to mid-February, and the hot season is from mid-February to the beginning of May [23]. The mean annual precipitation in the Chao Phraya River basin is 1487.4 mm, 90% of which falls during the rainy season [24]. The main river from the confluence of four upstream waterways to the river mouth is 396 km long, which can be divided into three sections in terms of river classification based on water quality standards: lower (river km (RKM) 7 to 62), middle (RKM 62 to 142), and upper (RKM 142 to 379) [25]. Up to now, flood disasters have been the central theme in studies related to the river, ranging from modeling to damage estimation [26–32]. The water quality of the river has been studied much less than floods and water resources management, although the pollution is serious [33,34]. The socioeconomic impacts of the major flood disasters that have occurred in the watershed may have affected the research priority setting for this river.

Water quality standards in Thailand were established in 1994 and the surface water quality standards are classified into five classes, as shown in Table 1 [35]. The water quality standards are centered on dissolved oxygen (DO), biochemical oxygen demand, and total coliform bacteria.

**Table 1.** Water quality standard of Thailand.

| Water Quality Parameter | Standard Value for Different Class | | | | |
|---|---|---|---|---|---|
| | Class 1 | Class 2 | Class 3 | Class 4 | Class 5 |
| Dissolved Oxygen (mg/L) | Natural | >6 | >4 | >2 | Very poor, |
| Biochemical Oxygen Demand (mg/L) | Natural | <1.5 | <2 | <4 | not classified in |
| Total Coliform Bacterial (MPN/100 mL) | Natural | <5000 | <20,000 | _ | class 1–4 |

The Thai government's Pollution Control Department (PCD) has been monitoring the water quality of Thailand's major rivers since the 1980s, according to our interview with Thailand government officials. Monitoring data from the period 1984–1995 showed that the water quality in the upper region of the Chao Phraya River was better than in the middle and lower reaches, as shown in Table 2 [25,36].

**Table 2.** Classification of Chao Phraya River by Thailand standards.

| Distance from River Mouth (RKM) | River Classification (Class) |
|:---:|:---:|
| 7–62 | Class 4 |
| 62–142 | Class 3 |
| 142–379 | Class 2 |

Monitoring results from the period 1993–2003 also identified the four most heavily polluted waterways in Thailand: the lower Chao Phraya River, the lower Ta Chin River, the lower Lam Ta Kong River, and Songkhla Lake [37]. The lower part of the Chao Phraya River was seriously polluted, especially from organic contamination due largely to household wastewater. The most alarming water quality problem in this part of the river was low concentrations of DO during the dry period from November to April. DO levels of most monitoring stations were lower than the regulated value in the established water quality standard for industrial use (not less than 2 mg/L). The average DO value in this section from 1978 to 1999 was 1.7 mg/L and the P20 (20th percentile) value during that period of time was only 0.5 mg/L [25].

Although distinct spatial differences in water quality along the Chao Phraya River were identified and found to be highly related to anthropogenic stresses stemming from communities, industries, and agriculture [38], the existing data were still insufficient to provide the details of the spatial and temporal variations in the water quality of the river. In particular, the ways in which the characteristics of water quality in the river course change with the change in season and within a season are still poorly understood. Furthermore, considering the fact that the population in Bangkok has been steadily increasing and there is a large variation in population distribution within the watershed, another research question is how the population distribution is linked to spatial variations in water quality.

*2.2. Methods*

In the present study, a multi-angle approach was employed to provide the characterization of water quality in the river. This approach assessed the water quality in terms of chemical parameters of water quality, bacteria, and phytoplankton composition. It was also intended to examine the relationship between populations, channel morphology, and water quality. Six water quality measurements were conducted along the Chao Phraya River between 2016 and 2018. Two of them were conducted in December and March, in the dry season. Two were conducted in August and September, in the wet season, and another two were undertaken in May, in the transition period between the wet and dry seasons. The number of sampling sites in these surveys ranged from 15 to 28 along the waterway from Nakhon Sawan, the starting point of the main Chao Phraya River, to a downstream point 16 km from the river mouth. Particular attention was placed on the longitudinal and lateral variations of water quality in two river bends located in highly populated lower reaches of the river course. The measured water quality parameters included water temperature, pH, chemical oxygen demand (COD), dissolved oxygen (DO), electrical conductivity (EC), nitrate ($NO_3$-N), ammonium ($NH_4$-N), orthophosphate ($PO_4$-P), and *E. coli*.

The present study adopted a purposive sampling method in both space and time for water quality investigation along the Chao Phraya River. Purposive sampling is a widely used sampling technique in which targets are selected according to pre-determined criteria based on the research objectives. By choosing the right information-rich targets, it can achieve the most effective use of limited resources [39,40]. Since our focus was the relationship between spatial variation and population distribution, as well as the effect of river channel morphology on water quality, sampling locations, as shown in Figure 1, were mainly concentrated in four areas: (1) Nakhon Sawan, which is the starting point of the main Chao Phraya River; (2) Ayutaya, an ancient and sizable city in the middle reaches of the river basin whose population has been increasing over the last five years, according to the National Statistical Office of Thailand; (3) Koh Kret, which has both an oxbow bend and a straight short cut of the river; and (4) Bang Krachao, which is located in the downstream section of the river in the heart of

Bangkok City, where the river course becomes a large bend. In the time axis, samplings were carried out in the dry season, at the beginning of the rainy season, and during the second half of the rainy season. On each sampling day, samples were taken between 10:00 am and 16:00 pm, as our pre-test survey on normal days in 2016 found insignificant diurnal variation of water quality in the river.

DO and its saturation level were measured using a Hach LDO (Luminescent Dissolved Oxygen) Probe, produced by Hach Company, Loveland, CO, USA. pH, EC, and $NO_3$-N were measured using a Horiba LAQUAtwin meter, made by HORIBA, Ltd., Kyoto, Japan. COD, $NH_4$-N, and $PO_4$-P were measured using a portable multi-parameter water analyzer manufactured by Kyoritsu Chemical-check Lab. Corp., Tokyo, Japan.

COD is similar in function to biological oxygen demand (BOD), in that both measure the number of organic compounds in water. In the management of Japan's rivers, BOD is more frequently used than COD for assessing river water pollution. Considering the fact that BOD reflects the oxygen demand of microorganisms oxidizing organic matter in the water under aerobic conditions, and the dissolved oxygen level in the Chao Phraya River is seriously low, especially in the downstream, the aerobic breakdown process may be affected or even inhibited by low dissolved oxygen. Therefore, using COD may be considered a better choice than BOD for the Chao Phraya River, since COD testing utilizes an oxidizing reagent to chemically oxidize all the organic and inorganic pollutants in the water. In addition, ammonium is high in BOD tests, which usually measure dissolved change for a five-day period—a period chosen by the Royal Commission on Sewage Disposal (UK) in 1908 according to the retention time of the Thames River at the tidal zone. The annual mean discharge of the Chao Phraya River is 700–800 m$^3$/s [41], which makes the retention time of the Chao Phraya River much longer than five days. Therefore, the hypothesis that most of the organic pollutants will be oxidized within five days may become invalid for the Chao Phraya River. Experimentally, the incubation period can be extended to 10 days or even longer, however, there is no consensus about this at present. Standard Methods for the Examination of Water and Wastewater (21st Edition) from the American Public Health Association (APHA) recommends an incubation time of 60 days [42], which seems impractical. Following these considerations, the present study measured only COD.

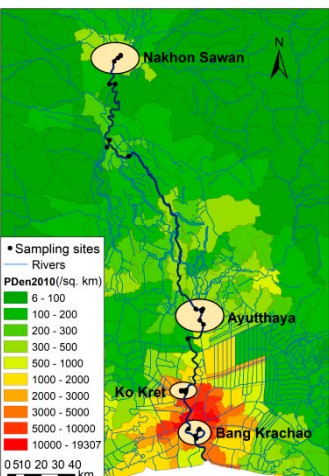

**Figure 1.** Four targeted river reaches for survey in the present study (PDen2010: Population Density in Thailand in 2010. Source: National Statistical Office, Thailand).

The *E. coli* detection was conducted using MicroSnap™ *E. coli* Test Kit, which is a rapid bioluminogenic method for detection and enumeration of *E. coli* bacteria. Samples taken from the survey sites were incubated in growth media for six hours. The detection device was then activated, and samples were further incubated for 10 min. At this time, a specific substrate reacts with diagnostic enzymes to produce light. The light generating signal is then quantified in the EnSURE luminometer. Both the Test Kit and EnSURE luminometer are manufactured by Hygiena, Camarillo CA, USA.

Phytoplankton are free-floating, photosynthesizing microscopic biotic organisms with a size range of 20–2000 μm. Phytoplankton account for approximately 50% of the global primary production, providing a food source for higher order organisms such as zooplankton and small fish. Since phytoplankton respond rapidly to changes within the surrounding environment, they may serve as an important biomarker for assessing the quality of water, as well as being an indicator of water pollution [43].

The phytoplankton survey was conducted only around the beginning of the wet season in 2018, for the purpose of shedding some light on the algae communities in the river and how the alga composition might be affected by a river bend, which is an important morphological feature of rivers. Since information on the phytoplankton composition in the river was not previously available, the pioneering, although one-time, survey carried out between the two seasons can be considered valuable and enlightening. Such a judgment-based sampling allows us to achieve depth of understanding, although breadth of understanding will be the aim of further studies.

Seventeen sites were sampled, from the middle stream of the Chao Phraya River to a downstream point 16 km from the river mouth. For every 1 L of water sample collected, 15 mL of Lugol's solution was added for plankton preservation. Lugol's solution is a harmless solution made from potassium iodide and iodine, which can add weight to cells to facilitate plankton settling, and stains cells a dark brown color to make identification and counting easier.

A Leica DM 750 Microscope (product of Leica Microsystems, Tokyo, Japan) was used for the identification and enumeration of phytoplankton species. Identification of phytoplankton species was based on morphological and other visible criteria, with reference to key literature and websites for phytoplankton identification. A plankton chamber with 400 grid boxes with a length of 0.5 mm was used for cell counting. The 10 times eyepiece and 20 times objective lens were used, allowing images with 200 times magnification to be obtained so that phytoplankton could be clearly observed and counted.

Mann–Whitney U and Kruskal Wallis tests were used to identify significant differences ($p < 0.05$) in water quality between bend and straight channel.

It should also be mentioned here that the main focus of this paper was on nutrients, *E. coli*, and phytoplankton communities in the river course, as less information of this type already exists. Moreover, the watershed population distribution map was used as the base map for all figures presented in the paper. Overlaying such information clearly shows how many people are living with degraded water.

## 3. Results

### 3.1. Overall Patterns of Spatial–Temporal Variation

Overall, the spatial variations in water quality, in terms of chemical parameters in the longitudinal direction, were similar to those reported in the 1990s. Water quality in the upper and middle reaches of the Chao Phraya River was better than in the lower reaches. Low DO concentrations in the lower reaches, as shown in Figure 2, remain a particularly serious concern. Although DO was higher in the wet season than the dry season, it was still under-saturated, as shown in Figure 3.

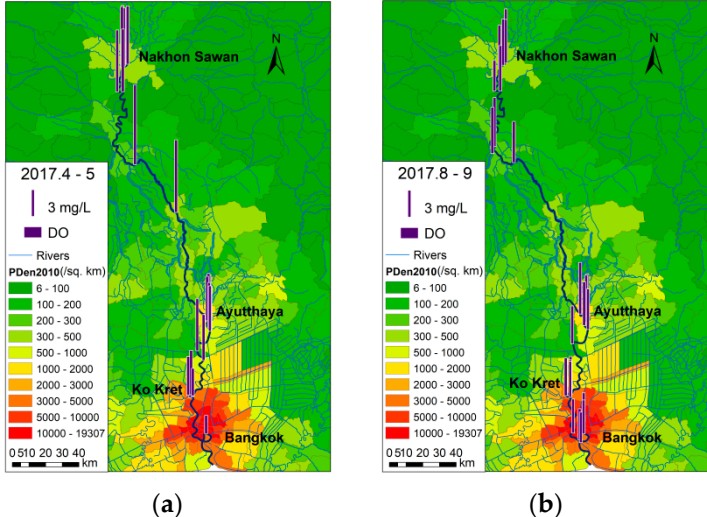

**Figure 2.** DO distributions at the beginning (**a**) and in the second half (**b**) of the wet season.

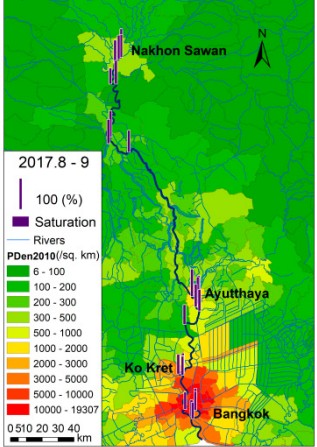

**Figure 3.** DO saturation.

With regard to nutrient levels, previous studies [44–46] have reported the correlation between flow discharge and nitrate concentration. For nitrate in particular, since it is highly soluble, it may be expected that the concentration increases as water flow increases. However, the present study found that nitrate concentrations were high in the dry season and also at the beginning of the wet season. At one downstream site, the nitrate concentration was very high at the beginning of the wet season, but decreased during the latter half of the wet season (Figure 4). This suggests that flood waters have a diluting function, reducing nitrate concentration by the end of the wet season. The study by Thimakorn also reported high concentrations of nitrate at the river mouth, and considered nitrate to be the most significant factor in causing oxygen depletion [47]. The present study confirmed the occurrence of high nitrate in the downstream reach, but showed that nitrate was not correlated to low DO.

Concentrations of $NH_4$-N and $PO_4$-P along the river course in the latter half of the wet season in 2017 were much lower than concentrations at the beginning of the wet season in 2018 (Figures 5 and 6). This again implies the cleaning function of flood waters. It can also been observed from Figures 5 and 6 that both $NH_4$-N and $PO_4$-P concentrations were high in the downstream section in the dry season of 2016.

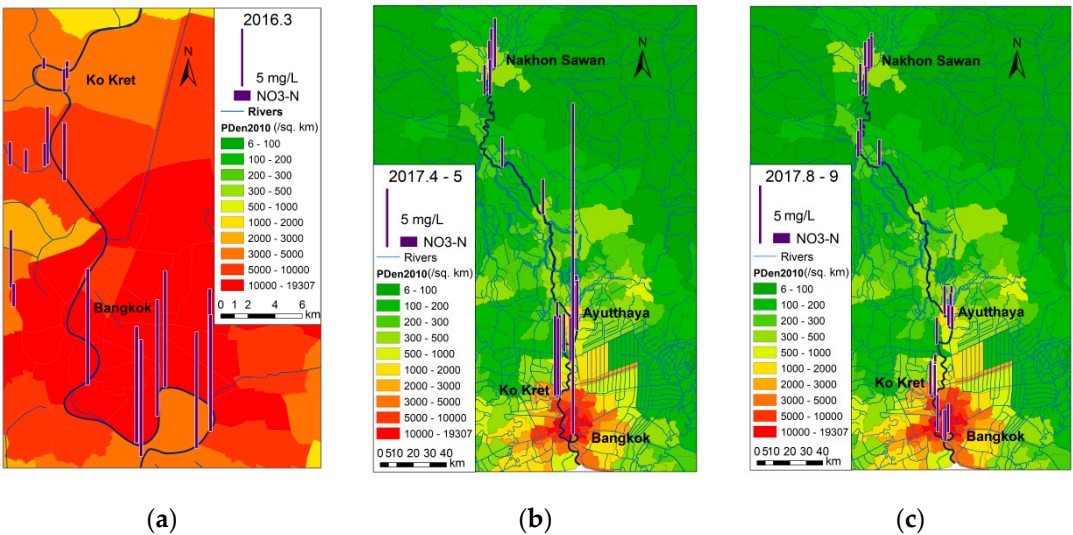

**Figure 4.** The spatial–temporal variation of NO$_3$-N. (**a**) Dry season; (**b**) Beginning of the wet season; (**c**) Second half of the wet season.

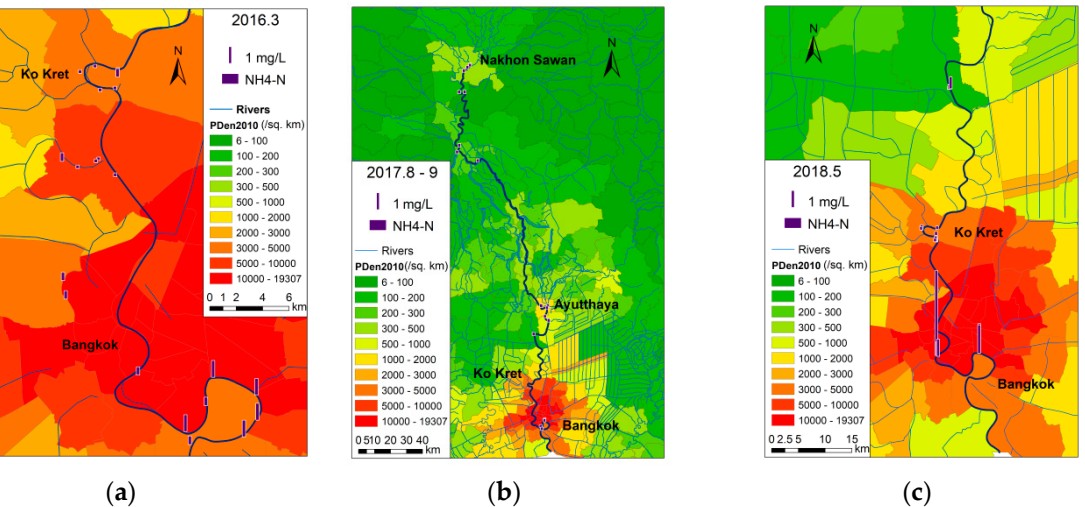

**Figure 5.** The spatial–temporal variation of NH$_4$-N. (**a**) Dry; (**b**) Beginning; (**c**) Second half.

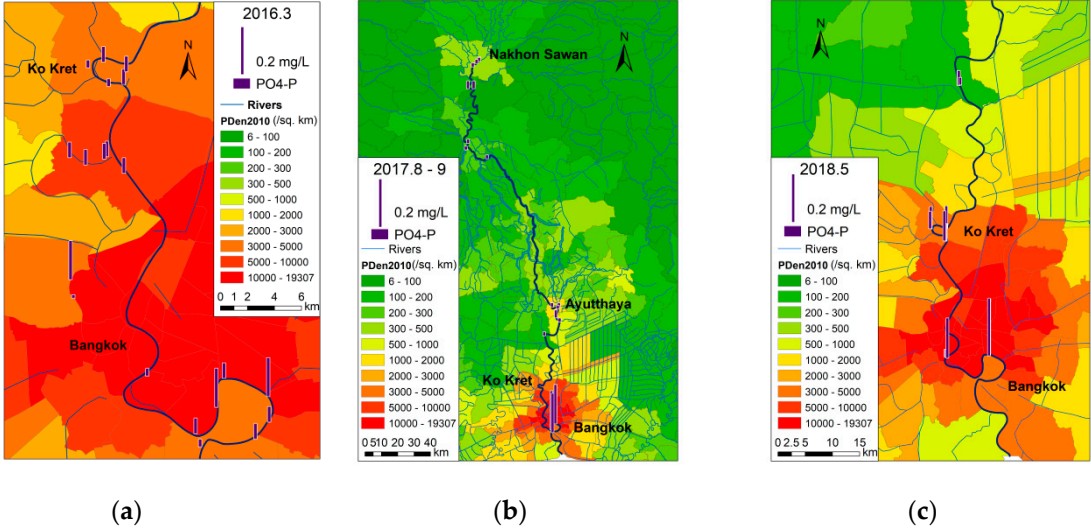

**Figure 6.** The spatial–temporal variation of PO$_4$-P. (**a**) Dry; (**b**) Beginning; (**c**) Second half.

All measured concentrations of COD ranged from 5 mg/L to more than 10 mg/L. Regardless of season, COD appears to be correlated with population density. pH values were in the range of 7.2–8.3 and the maximum measured EC was about 260 μS/cm. The data obtained in the second half of the rainy season in 2017 are shown in Figure 7.

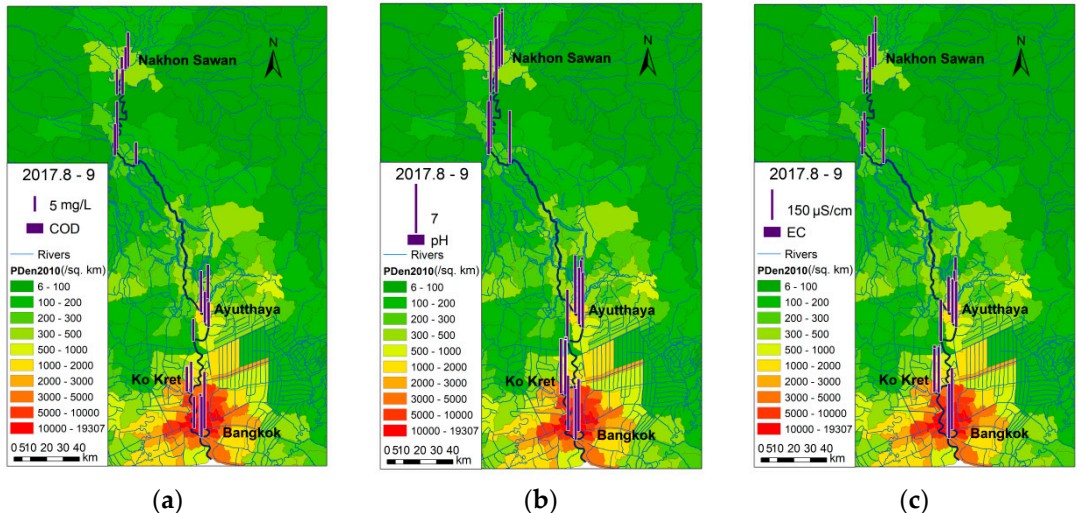

**Figure 7.** The spatial variations of COD (**a**), pH (**b**), and EC (**c**).

The spatial variation pattern of *E. coli* in the wet season was sag-curved, as depicted in Figure 8. It was higher in the upper and lower parts than in the middle. The reason for this is that there are inadequate and/or a lack of waste water collection and treatment facilities along the Ping and Nan rivers—the headstreams of the Chao Phraya River. The Chiang Mai Metropolitan Area has a population of nearly one million people and has been attracting over five million visitors each year. Therefore, during the wet season, large amounts of waste water could be transported into the waterways by overland surface runoff. It was reported that the total coliform bacteria at the Ratanakosin Bridge and Kanjanapisek Garden in Chiang Mai exceeds 1.6 million MPN/100 mL [48], which is at least 320 times the regulated standard surface water quality in Thailand. In the upstream section, it was observed that some public toilets were even placed on the river bank, and waste water was directly discharged into the waterway. It can also clearly be seen in Figure 8 that the less populated upstream area has a problem with *E. coli* of the same magnitude as the much more populated downstream area, which is caused by contamination dispersion from headwater regions. In Bangkok, the high level of *E. coli* contamination was caused by the large population, as evidenced by the fact that the levels of *E. coli* upstream of Bangkok were very low. The disproportionality between population and contamination is an issue of environmental ethics. More rigorous river management should be implemented in the Chiang Mai Metropolitan Area for the well-being of residents in both the Chiang Mai Metropolitan Area and Nakhon Sawan Province.

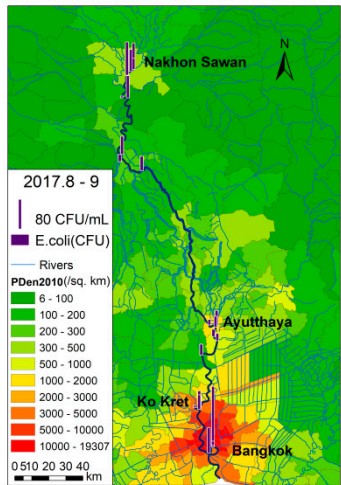

**Figure 8.** The distribution of *E. coli* in the latter half of the rainy season.

During the dry season, however, the level of *E. coli* was found to be much lower; approximately 10 CFU/mL at our sampling sites. According to the meteorological information for Thailand, sun hours are higher during the dry season than the rainy season. In addition, cloudiness is low during the dry season (https://www.timeanddate.com/weather/thailand/bangkok/climate). Since the effect of sunlight exposure on pathogen survival has been well documented, the low levels of *E. coli* present during the dry season may be explained by the strong UV radiation occurring during the dry season, which may inhibit *E. coli* growth.

Figure 9 shows that there was a high spike in *E. coli* levels (880 CFU/mL) at the beginning of the rainy season in 2018, at a downstream site located in the most populated area. It is similar to the first flush effect for nutrients in storm water runoff from catchment. However, the rainy season did not manifest the cleaning effect in the latter half of the season.

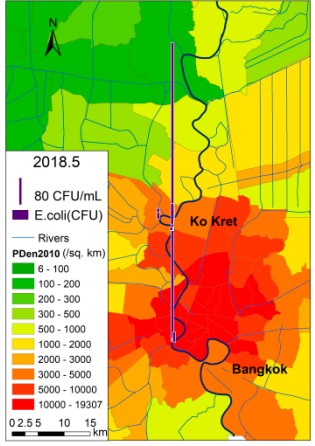

**Figure 9.** The spike of *E. coli* at a downstream site.

*3.2. Spatial–Temporal Variation along River Bends*

Koh Kret (sometimes spelled Koh Kred or Ko Kret) in Bangkok, where the natural landscape dominates and concrete paths are too narrow for cars, is an artificial island that was cut off from the city by an 18th-century canal, a shortcut in a bend in the Chao Phraya River, as shown in Figure 10a. In the Ayutthaya Period, when boats were the major mode of transportation between the old capital and the lower parts of the river, as well as the Gulf of Siam (now the Gulf of Thailand), travelling along this winding section of the waterway took a long time, so in 1722 King Thai Sa ordered the construction of the shortcut, shortening the distance by 75% [49]. Unlike many other shortcuts in

waterways around the world, the bend was not turned into land. Therefore, this is a good site for studying how the water quality in a bend may be different from that in a shortcut.

As an island, Koh Kret can currently only be accessed by boat. However, this constraint turns out to be of great advantage to the community, resulting in little development, a unique riparian landscape (Figure 10b), and the harmony of the ethnic minority Mon and Thai groups living and sharing cultures and traditions in the same place [48]. Due to its unique physical, social, and cultural virtues, Koh Kret has become an eco-cultural tourist attraction in Thailand.

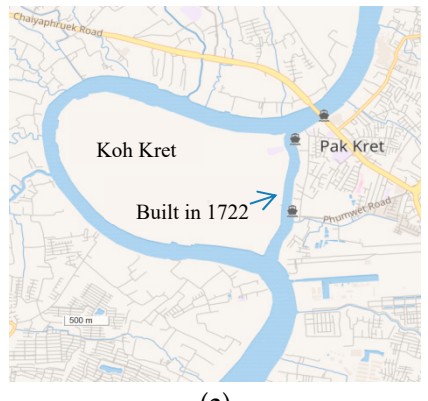 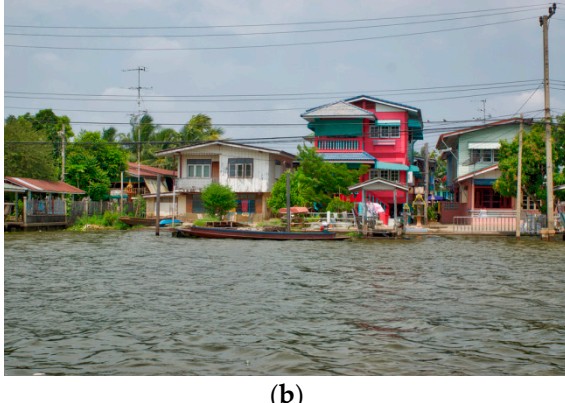

(**a**) (**b**)

**Figure 10.** River landform at Koh Kret and a typical riparian landscape along the bend and shortcut. (**a**) River landform; (**b**) Typical riparian landscape.

Downstream of Koh Kret, the Khlong Lat Pho Floodgate Project led to the construction of a shortcut through the river bend at Bang Krachao. This shortened the length of the river bend from 18 km to only 600 m. The shortcut canal is gated on normal days and open when there is a large flood. Water quality measurements were conducted along the two bends and the straight canal at Koh Kret. However, the Klong Lad Pho canal was not surveyed, due to its closure.

For chemical parameters of water quality, statistically significant differences in concentrations between the bend and straight canal were not found. However, *E. coli* levels behaved differently. At the beginning of the wet season in 2018, the levels of *E. coli* in the bend of Koh Kret were lower than in the straight canal (Figure 11a), and *E. coli* was not detected at all along the Bang Krachao bend. However, a large spike (880 CFU/mL) was detected at a location upstream of the bend. In the second half of the wet season in 2017, the levels of *E. coli* in the bend of Koh Kret were also lower than in the straight canal. It was also found that the level of *E. coli* in the Bang Krachao bend varied significantly, with low levels on the right bank side and high levels on the left bank side (Figure 11b). Since the largest slum in Bangkok is situated next to the left bank, the high level of *E. coli* on the left bank side may be attributed to waste water discharge from the slum.

The sampling locations for the phytoplankton survey and the number of species identified at each location are shown in Figure 12a. Out of all the survey sites, the number of phytoplankton species was the highest on the left side bank of the Bang Krachao bend (Figure 12b).

Seven species were found in 16 of the 17 survey sites. These were: (1) *Arthrospira platensis*, (2) *Oscillatoria sp.*, (3) *Oscillatoria tenuis*, (4) *Aulacoseira islandica*, (5) *Cyclotella meneghiniana*, (6) *Cyclotella sp.*, (7) *Scenedesmus acuminatus*.

*Arthrospira platensis*, also known as Spirulina, is a gram negative, non-toxic species of cyanobacteria, known across the world for its potential nutritional value. The most important factor that governs its growth is the presence of light. Although it is non-toxic, its excessive growth will produce an unattractive surface scum. Nevertheless, it is a very efficient producer of oxygen and, unlike other green algae, does not consume oxygen during the night [50]. Therefore, if this type of blue-green algae were removed from the river, oxygen levels would drop further with the potential for major fish kill.

*Oscillatoria* is a genus of filamentous cyanobacteria common in freshwater environments, deriving its name from its slow oscillating movement, and comprising more than 100 species. Due to its capability to slide back and forth, *Oscillatoria* can orient itself towards a light source so that it can grow under less favorable light conditions. In addition, it can produce both anatoxin-a and microcystins. *Oscillatoria tenuis* simultaneously produces geosmin and 2-methylisoborneol (MIB), which account for many odor problems in freshwater water [51–53]. The *Aulacoseiraceae* family is one of the oldest freshwater diatom families, long and filamentous in shape [54]. *Cyclotella meneghiniana* is also a large diatom with a diameter of 5–43 µm. *Cyclotella sp.* is a small, centric diatom with cells only 3–5 µm in diameter that may be used as important indicators of environmental change. Unlike most of the colonial green algae that form long filaments, *Scenedesmus acuminatus* is a green alga with small chains of four cells, with the potential to be used for biodiesel production.

At the river bend of Koh Kret, the dominant species was found to be *Oscillatoria tenuis*, which is an odor-producing species. In the straight canal, however, the phytoplankton community was dominated by *Botryococcus braunii*—a green, pyramid-shaped planktonic microalga. Blooms of *Botryococcus braunii* have been shown to be toxic to other micro-organisms and fishes. However, *Botryococcus braunii* is regarded as a potential source of renewable fuel because of its ability to produce large amounts of hydrocarbons [55]. The use of algal hydrocarbons can greatly reduce the environmental impact associated with using coal and petroleum. Also, a study suggests that a metabolic active biomass of *B. braunii* could be used for copper removal from solutions, while it produces appreciable quantities of hydrocarbons.

It was also found that, out of the 17 survey sites, *Cymbella affinis* was present only at the bend of Koh Kret. As a saproxenous species (algae found in clean water), *Cymbella affinis* can be used as an indicator of river pollution [56,57]. Therefore, the bend of Koh Kret may be characterized as having a coexistence of saproxenous and odor-producing species. Further in-depth study of such coexistence is needed for better river management.

Figure 13 presents a zoomed-in view of nutrients along the two river bends measured during the latter half of the rainy season in 2018. Nitrate concentrations were more or less the same along the two bends, while the highest ammonium concentration occurred where the spike in *E. coli* levels was detected. It can also been noted that phosphate concentrations were high in the Bang Krachao bend, where the number of phytoplankton species identified was the highest out of all the survey sites.

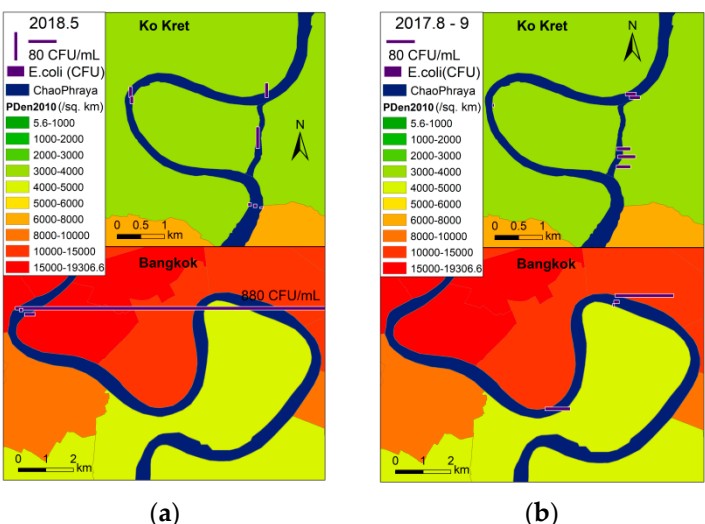

(**a**)                     (**b**)

**Figure 11.** The zoomed-in view of *E. coli* distribution along the two river bends. (**a**) At the beginning of the wet season; (**b**) In the second half of the wet season.

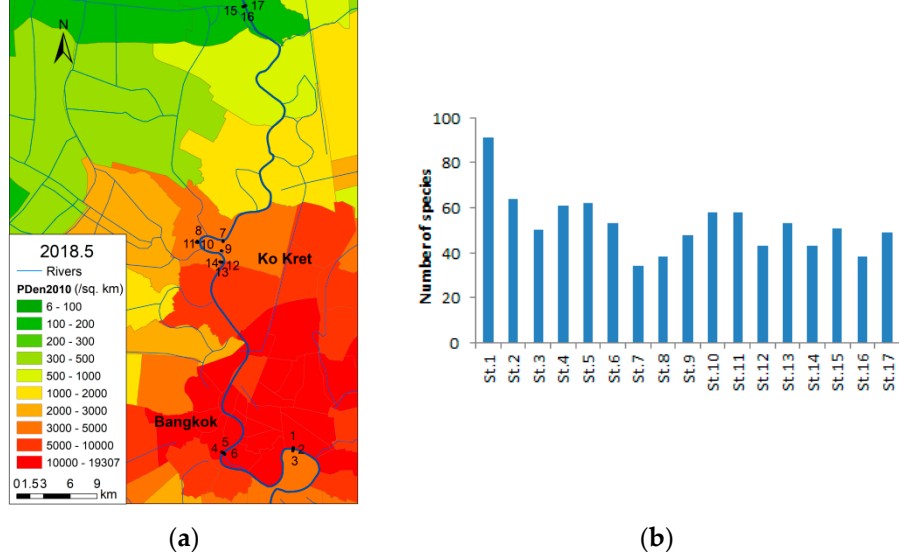

**Figure 12.** Sampling locations (**a**) and the number of identified species at each site (**b**).

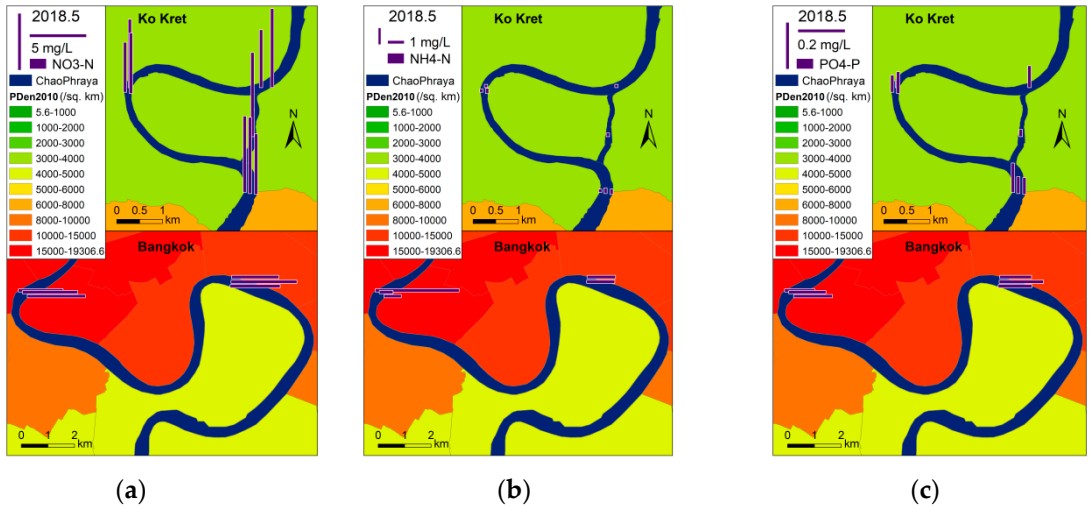

**Figure 13.** Nutrients concentrations along the two river bends. (**a**) NO$_3$-N; (**b**) NH$_4$-N; (**c**) PO$_4$-P.

## 4. Discussion

The first flush phenomenon has been reported in previous studies [58,59] however, these studies were focused on a storm event. Therefore, a first flush usually refers to the initial surface runoff of a rainstorm, when the first runoff has a high concentration relative to runoff later in the storm event. The findings of the present study suggest that this concept may also be applied to a rainy season, which can be termed as a seasonal first flush. More importantly, the present study proposed a new concept of a duality of seasonal effect. Expanding the notion of the first flush in such a way highlights both the negative and positive roles of the rainy season, which may lead to a better understanding of river dynamics and better river management as well. Additionally, the present study also indicated that such a duality was not observed for *E. coli*. This implies a continuous loading of *E. coli* from the watershed to the river during rainy season. Because the flood risk along the Chao Phraya River is high, the *E. coli*-contaminated flood waters coming out the river channel will cause not only economic damage but also health damage. Since Bangkok is one of the world's top tourist destinations, and the boat trip along the Chao Phraya river is a popular attraction for visitors, a high level of *E. coli* in the river can be considered a health risk to tourists. Despite this, the flood-related health issues have been largely overlooked up to now. On the other hand, since the number of visitors to Thailand peaks in

August, as shown in Figure 14, the high level of *E. coli* in the rainy season may be partially attributed to tourists. Therefore, tourists in Bangkok could be both the victims of *E. coli* and one of the causes of the contamination.

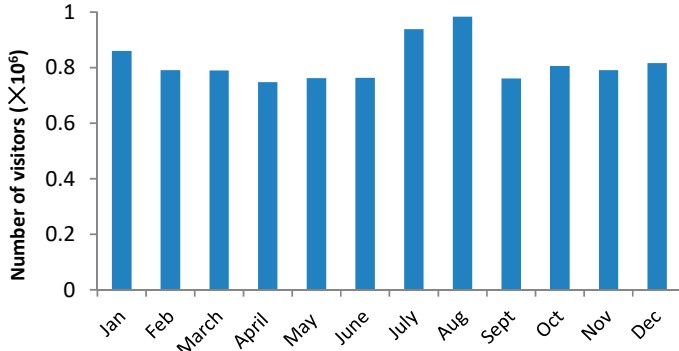

**Figure 14.** Monthly variation in visitors to Thailand in 2017 (Source: Department of Tourism, Thailand).

Furthermore, as shown by the present study, the high level of *E. coli* on the left bank side of Bang Krachao can also be considered to be partially related to Khlong Toey, one of the largest low-income communities in Bangkok. Therefore, eliminating the health risk is not simply a matter of sewer construction, but also requires an innovative solution.

For the relationship between water quality and river bend, field data suggested another type of duality in relation to river bend. Although a river bend may be effective in reducing *E. coli*, it may also host harmful algae. In addition, the fact that the number of phytoplankton species in the bend of Bang Krachao (which is located in an urbanized and populous area) was significantly higher than all upstream sites is proof that our understanding of river ecosystems is still very much limited.

Studies on river meanders have tended to focus on the ecological functions of river bends, and discussions on the possible negative aspects of river bends are missing in scientific literature. The present study serves as a call for more comprehensive studies on river meanders, both ecologically and socially.

A water quality index (WQI) is a means by which water quality data are combined to produce a score in a consistent manner, which informs the public and decision-makers of the suitability of water for various uses. This concept was developed in 1965 [60] and led to the development of various methods of WQI calculation, such as the NSF-WQI proposed by the National Sanitation Foundation of the United States [61], and the CCME WQI proposed by the Canadian Council of Ministers of the Environment [62]. The most commonly used parameters in calculating a WQI are dissolved oxygen, pH, turbidity, total dissolved solids, nitrates, phosphates, and metals, among others. Since the present study was intended to investigate the spatial–temporal variation in each individual water quality parameter, a WQI was not employed. However, it is worth pointing out here that various water quality indexes have been calculated based on chemical or biological parameters. As evidenced by Koh Kret, a river reach with a water-mediated landscape and water-nourished community can provide recreational and cultural values, which has not been given any consideration in existing WQIs. Therefore, further efforts to combine water chemistry, bacteria, algae, and river scape to develop a more integrative WQI should be pursued. In addition, one of the major shortcomings in many existing WQIs is that the interrelationships between the variables used in an index are usually ignored. When phytoplankton is involved, its growth may deplete nutrients. Consequently, the consideration of interrelationships between variables becomes indispensable, although methods for dealing with such interrelationships are yet to be established.

## 5. Conclusions

The present study revealed a duality of the rainy season with regard to water quality. High water may cause water quality deterioration around the beginning of the rainy season, but improve water quality via dilution in the end.

For *E. coli*, the situation is different. While nutrient concentrations were diluted during the latter half of the rainy season, levels of *E. coli* remained high. During the dry season, however, the levels of *E. coli* were suppressed, probably owing to the high levels of solar radiation reaching the water surface. In addition, the spatial variation of *E. coli* along the main Chao Phraya River was found to be a sag curve, with high levels in the upstream and downstream.

The comparative observation between bend and straight river reaches found no significant differences in the chemical parameters of water quality. *Oscillatoria tenuis*, a harmful odor-producing alga, was identified as the dominant phytoplankton species at the bend. However, a saproxenous diatom species was found to coexist with *Oscillatoria tenuis* at the bend. In the straight canal, *Botryococcus braunii*, which can be used to produce biofuel, was dominant. The lesson learned here is that, although a river bend is an important ecological habitat, it may also be utilized as a breeding ground by harmful species. On the other hand, as river restoration projects aiming to increase habitat heterogeneity by re-meandering straightened reaches have gained momentum because straight channels are perceived as being of low ecological value, the findings of the present study indicated that straight channels may have bioenergy-related beneficial values, which have not been previously recognized. The present study also suggests that the presence of *Arthrospira platensis* in the waterway can be considered beneficial due to its oxygen production in this oxygen-deficient river. Finally, it points out a new direction for further water quality index development.

**Author Contributions:** G.H. designed this study, carried out all surveys together with other authors, and prepared this manuscript. H.X. and H.L. participated in all field works and data processing and H.L. was also involved in phytoplankton identification and enumeration. C.E. and T.S. contributed to this study by joining field works, advising on sampling site selection, some instrument preparation and logistics as well.

**Funding:** This research was funded by Ministry of Education, Culture, Sports, Science and Technology-Japan under the name of Sophia Research Branding Project.

**Acknowledgments:** Thanks should be given to Bai Fan, Yu Wei, and Liu Shizhou for their assistance in phytoplankton analysis.

**Conflicts of Interest:** The authors declare no conflict of interest.

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
