# Peer review of "Duality of Seasonal Effect and River Bend in Relation to Water Quality in the Chao Phraya River"

_water, doi:10.3390/w11040656_

Round 1

Reviewer 1 Report

Recommendations:

- Explain more clearly the way the results are analyzed;

- The equipment and the methods for determination of physicochemical parameters are not specified;

- The results of pH, temperature, conductivity, BOD monitoring are not presented;

- In the Materials and Methods chapter it is stated that the study was carried out between 2016 and 2018. The results obtained during 2016 do not appear.

- Water quality classification is usually done by calculating quality indices such as water quality index (WQI), water pollution index, saprobe indicator, etc. If it is desired to classify water in quality classes it would be good to apply one of these methods;

- Enrich the bibliographic references and updated with recent studies.

Author Response

Dear Reviewer

Thank you for your professional comments. It helped a lot, really appreciated.

Guangwei Huang

Reviewer 2 Report

Dear authors,

please, find attached the annotated manuscript with extensive observations.

At present the manuscript needs a lot of work to be suitable for publication.

The most important thing is thinking in the objectives of the paper and if the data presented in the manuscript are suitable for accomplishing the objectives. It is hard to judge it from the methodology section because there is not enough information about the choice of sampling points, location (there are no maps), frequency and parameters. Why COD and not BOD which is the parameter included in Thailand legislation? Why phytoplanton in just one survey? How can we be able to analyze "variation in relation to river bend" if the authors don't provide a detailed sampling design? I mean spatial and temporal.

Also, the references included are very scarce, plenty of references needed (pointed in the annotated pdf). No references in the methodology section. 

In summary, at present state I cannot judge if design and methodology were adequate for the objectives.

Author Response

Dear Reviewer

Thank you for your professional comments. They are very helpful, really appreciated.

Guangwei Huang

Round 2

Reviewer 1 Report

Improvements have been made to the content. For future studies, you should also consider interpreting from the point of view of quality indices. Only so will you have conclusive results.

Author Response

Dear Reviewer

Thanks again for your professional advice. It does not only help improve the present work, but also provides hint on further study. Really appreciated.

Additional explaination on WQI was added.

Best regards

Guangwei Huang

Reviewer 2 Report

In my first review I took the time to made quite a few suggestions directly on the paper, using Acrobat comments tool.

I have checked the updated version and the authors did not take them into account. They only answered the specific comments written here.

I would appreciate that authors take the time to see the annotated pdf file, maybe they did not see it or if they do not agree with my observations at least provide an answer.

I attach again the anotated file

Author Response

Dear Reviewer

First of all, sorry for missing your comments directly embedded in the file. They are so helpful in improving the present work. Really apprecite your time and efforts in maintaining high quality of academic publication. In the second revision, all your comments were followed. References were added and ambiguous parts were clarified.  

Thank you again for your professional advice.

Best regards

Guangwei Huang

Round 3

Reviewer 2 Report

I appreciate the authors’ effort to try to improve their manuscript. However, I think that they are doing minimum changes to try to accomplish. Maybe, the time for their revision is not enough and they need more time to solve it adequately.

The paper has serious structure problems. The authors mix introduction and material and methods section. It is completely necessary to rearrange both sections.

Introduction should focus first on describing the state of art about the two topics they are working 1) seasonal and spatial variability of river water quality and 2) relationship between water quality and river morphology. The analysis can be global (all earth), or more specific to similar areas (e.g. those suffering monsoon effects) or even specific to Thailand (in EMAS journal accepts regional studies). Regarding this I find some sentences that are completely inacceptable, without a deep review of current knowledge. For example:

1) Line 27 – 29 “Although river water quality has been studied either routinely by river management authorities (…) our understanding of spatial-temporal variations of water quality is still far from perfect.” One sentence cannot substitute an adequate review.

2) Reference [2] it is not appropriate for justifying above paragraph. There is plenty of research

3) “Our literature search using Web of 40 Science with the keyword of “Chao Phraya River flood” had 33 hits while the keyword of “Chao 41 Phraya River water quality” led to only two publications.” This is not appropriate at all. Authors should read that references and extract relevant information

4) Line 43 to 45 “(…) lower (river km (RKM) 7 to 62), middle (RKM 62 to 142) and upper (RKM 142 to 379).” Reference still needed or locate that paragraph after Table 2, where it is more appropriate. Moreover, this information should be included in a subsection named “study area”

5) line 70 first reference to Bangkok, if the authors aim at an international audience they should describe adequately the study area in an understandable way. A study area figure should be included from the first time they start talking about Chao Phraya River. Moreover, this information should be included in a subsection named “study area”

6) Line 77 – 79 “Although the interaction of flow regime and water quality has been studied previously [12], the relationship between river channel characteristics and water quality has been largely neglected up to now and is an important research question to be explored” I’m sorry but I’m sure that there is research about this topic. Make an effort to search it, and to specify which is the novelty of your study compared to previous research. Please, take in mind my previous observation because it is not the same in Europe or in North America or in Asia, etc. so, specify always if you are talking globally or of a specific area.

7)Line 100 to 111 “Six water quality measurements were 100 conducted along the Chao Phraya River between 2016 and 2018 (…)” This is “materials and methods” section

To sum up, I recommend to develop a State of art review first in the introduction. All specifications about study area should be included in a subsection “Study area”, explain why the authors chose to work with Chao Phraya river. State clearly the objectives based on the previous sections. Accommodate all the “material and methods” in the appropriate section, avoiding repetitions

Author Response

Dear Reviewer

Thank you again for your invaluble comments. Revisions were done accordingly, resulting in a much better manuascript.

Best regards

Guangwei Huang
